# EB1089 Increases the Antiproliferative Response of Lapatinib in Combination with Antiestrogens in HER2-Positive Breast Cancer Cells

**DOI:** 10.3390/ijms25063165

**Published:** 2024-03-09

**Authors:** Angèle Sorel Achounna, David Ordaz-Rosado, Janice García-Quiroz, Gabriela Morales-Guadarrama, Edgar Milo-Rocha, Fernando Larrea, Lorenza Díaz, Rocío García-Becerra

**Affiliations:** 1Departamento de Biología Molecular y Biotecnología, Instituto de Investigaciones Biomédicas, Universidad Nacional Autónoma de México, Ciudad de México 04510, Mexico; xachounna@gmail.com (A.S.A.); edgarmilorocha@iibiomedicas.unam.mx (E.M.-R.); 2Departamento de Biología de la Reproducción Dr. Carlos Gual Castro, Instituto Nacional de Ciencias Médicas y Nutrición Salvador Zubirán, Vasco de Quiroga No. 15, Belisario Domínguez Sección XVI, Tlalpan, Ciudad de México 14080, Mexico; david.ordazr@incmnsz.mx (D.O.-R.); janice.garciaq@incmnsz.mx (J.G.-Q.); gabriela.mguadarrama@gmail.com (G.M.-G.); fernando.larreag@incmnsz.mx (F.L.); lorenza.diazn@incmnsz.mx (L.D.); 3Programa de Investigación de Cáncer de Mama, Instituto de Investigaciones Biomédicas, Universidad Nacional Autónoma de México, Ciudad de México 04510, Mexico

**Keywords:** HER2-positive breast cancer, EB1089, lapatinib, tamoxifen, fulvestrant

## Abstract

HER2-positive breast cancer is associated with aggressive behavior and reduced survival rates. Calcitriol restores the antiproliferative activity of antiestrogens in estrogen receptor (ER)-negative breast cancer cells by re-expressing ERα. Furthermore, calcitriol and its analog, EB1089, enhance responses to standard anti-cancer drugs. Therefore, we aimed to investigate EB1089 effects when added to the combined treatment of lapatinib and antiestrogens on the proliferation of HER2-positive breast cancer cells. BT-474 (ER-positive/HER2-positive) and SK-BR-3 (ER-negative/HER2-positive) cells were pre-treated with EB1089 to modulate ER expression. Then, cells were treated with EB1089 in the presence of lapatinib with or without the antiestrogens, and proliferation, phosphorylation array assays, and Western blot analysis were performed. The results showed that EB1089 restored the antiproliferative response to antiestrogens in SK-BR-3 cells and improved the inhibitory effects of the combination of lapatinib with antiestrogens in the two cell lines. Moreover, EB1089, alone or combined, modulated ERα protein expression and reduced Akt phosphorylation in HER2-positive cells. EB1089 significantly enhanced the cell growth inhibitory effect of lapatinib combined with antiestrogens in HER2-positive breast cancer cells by modulating ERα expression and Akt phosphorylation suppression. These results highlight the potential of this therapeutic approach as a promising strategy for managing HER2-positive breast cancer.

## 1. Introduction

Breast cancer is the most common type of malignancy affecting women worldwide. Additionally, this disease continues to be the leading cause of cancer-related deaths among women [1].

In clinical practice, breast cancer tumors can be mainly classified into three molecular subtypes: estrogen receptor (ER) positive, human epidermal growth factor receptor 2 (HER2)-positive, and triple-negative [2]. HER2-positive breast cancer is known for its particularly aggressive nature, often resulting in reduced progression-free survival and overall survival rates. It comprises 20–30% of all breast cancers and is characterized by overexpression of the human epidermal growth factor receptor 2 (HER2, HER2/neu, ERBB2) oncogene. Among HER2-positive breast cancers, approximately 50% are also ER-positive, leading to the division of patients into two main subgroups: ER-negative/HER2-positive and ER-positive/HER2-positive, each with distinct growth patterns and responses to treatment [3].

For the first subgroup, ER-negative/HER2-positive, which does not respond to endocrine therapy due to the absence of ER, the treatment strategy typically involves chemotherapy alongside anti-HER2 drugs, such as the monoclonal antibody trastuzumab or the tyrosine kinase inhibitor (TKI) lapatinib. The latter suppresses the phosphorylation of both HER2 and EGFR, thus disrupting their downstream signaling pathways, such as MAPK and AKT. These pathways play pivotal roles in cell proliferation and survival [4,5]. On the other hand, various approaches are currently under investigation for the ER-positive/HER2-positive breast cancer subgroup, often involving antiestrogenic therapy combinations with anti-HER2 treatment [6]. However, not all patients benefit equally from these treatments, as some may not respond adequately or develop resistance to the drugs [7].

Calcitriol, the active metabolite of vitamin D, and some of its analogs, such as EB1089, increase the sensitivity of tumor cells to various chemotherapeutic agents, antiestrogen compounds, TKIs, and ionizing radiation [8,9,10,11]. Previous studies conducted in our laboratory demonstrated that calcitriol induced ERα expression in ER-negative breast cancer cells [12,13]. Consequently, calcitriol restored the growth-inhibiting potential of the antiestrogens tamoxifen and fulvestrant in both primary and established ER-negative breast cancer cells [12]. Considering all of the above, the main objective of our study was to investigate the effects of adding EB1089 to the combined treatment of lapatinib with antiestrogens in HER2-positive breast cancer. Specifically, we aimed to assess its impact on cell proliferation, the regulation of ERα, and the phosphorylation status of Akt and Src.

## 2. Results

### 2.1. EB1089 Inhibits Cell Growth and Modulates ERα Expression in HER2-Positive Breast Cancer Cells

Compared to calcitriol, EB1089 has demonstrated significantly greater potency (50–200 times) in regulating cell growth and differentiation across various cell types while exhibiting fewer hypercalcemic effects [14,15,16,17]. Therefore, we have selected this compound for our study.

Figure 1a depicts the effect of EB1089 on BT-474 cell growth, showing a concentration-dependent inhibition on cell proliferation. In particular, this effect was statistically significant, starting from the concentration of 1 × 10^−9^ M compared to untreated cells. The values of the inhibitory concentration at 20% (IC_20_) and 50% (IC_50_) were determined from the concentration-response curve of the compound and are presented in Table 1. Previous studies conducted by our laboratory have reported the IC values of EB1089 in SK-BR-3 cells and the IC values of lapatinib in both BT-474 and SK-BR-3 cell lines (Table 1) [9,10]. These values were employed in subsequent experiments.

Earlier investigations have shown that the effect of calcitriol on ERα expression varies depending on the initial status of the receptor expression. Specifically, in ER-positive breast cancer cells, calcitriol and its analog EB1089 exert antiproliferative effects, partly by downregulating ERα levels, leading to the attenuation of estrogen-like responses [18,19,20]. Conversely, calcitriol induces the expression of ERα in ER-negative breast cancer cells [12,13]. Herein, to assess the effect of EB1089 on ERα protein expression, BT-474 or SK-BR-3 cells were treated with two concentrations of the analog. As a control, the cells were also treated with calcitriol. In the BT-474 cell line, characterized by the expression of ER and HER2, both EB1089 and calcitriol reduced ERα expression at the two concentrations tested compared to the control (Figure 1b). In contrast, in the SK-BR-3 cells, which lack ER expression and overexpress HER2, EB1089 and calcitriol induced a dose-dependent upregulation of ERα expression (Figure 1c).

Our current findings indicate that EB1089 induces ERα expression in ER-negative breast cancer cells according to previous calcitriol studies [12,13]. Additionally, this corroborates earlier reports demonstrating that the calcitriol analog reduces ERα expression in ER-positive cells [20,21].
ijms-25-03165-t001_Table 1Table 1Inhibitory concentration values of EB1089 and lapatinib on the proliferation of HER2-positive breast cancer cells.CompoundICBT-474 (mol/L)SK-BR-3 (mol/L)EB1089202.2 × 10^−10^4.3 × 10^−10^ [9]505.6 × 10^−10^7.5 × 10^−10^ [9]Lapatinib20nd1.6 × 10^−8^ [9]502.5 × 10^−8^ [22]8.9 × 10^−8^ [9]Inhibitory concentration (IC), not determined (nd). The number in brackets corresponds to a reference.


### 2.2. EB1089 Enhances the Antiproliferative Effects of the Combined Treatment Comprising Lapatinib with Antiestrogens in HER2-Positive Breast Cancer Cells

In ER-positive/HER2-positive breast cancer cells, ER and HER2 are crucial in promoting cell proliferation and survival. Additionally, in ER-negative/HER2-positive breast cancer, resistance to HER2-targeted therapies can develop through the reactivation of the ER pathway as an escape mechanism [23]. Thus, simultaneously targeting ER and HER2 pathways, in both cases, is crucial for optimal treatment efficacy [23].

Previously, the enhancement of the antiproliferative activity of TKIs or antiestrogens by EB1089 in breast cancer cells has been demonstrated [9,24]. However, the simultaneous treatment with EB1089, lapatinib, and antiestrogens has not yet been studied. Therefore, we aimed to evaluate the effect of this combination on HER2-positive breast cancer cell proliferation. As part of our study, we included control groups to evaluate the effects of EB1089 in combination with either lapatinib or antiestrogens. BT-474 or SK-BR-3 breast cancer cells were pre-treated with EB1089 to regulate ERα expression. Subsequently, these cells were treated with lapatinib, the antiestrogens, either alone or in combination, with or without estradiol. After the treatment period, a proliferation assay was performed.

Figure 2a shows that estradiol (E, black bar) significantly increased BT-474 cell proliferation compared to vehicle-treated cells (C, black bar). In contrast, tamoxifen and fulvestrant significantly decreased cell growth compared to vehicle-treated cells, either alone (T and F, black bars) or combined with estradiol (E + T and E + F, black bars). Furthermore, the combined treatment of antiestrogens with estradiol (E + T and E + F, black bars) significantly reverted the proliferative estradiol effect (E, black bar). These results were expected since BT-474 cells express ERα and are thus responsive to these treatments.

In the control group (C) in Figure 2, the treatments of lapatinib or EB1089 alone (solid gray and striped black bars, respectively), as well as their combination (striped gray bar), significantly inhibited cell proliferation compared to untreated cells (C, black bar). Similar results were observed in the estradiol-treated group (E) when comparing the compounds combined with estradiol (solid gray, striped black, and striped gray bars) against estradiol alone (E, black bar). In particular, within this group, a significant reduction in cell growth was observed when EB1089 was added to the combination of lapatinib and estradiol (striped gray bar) compared to the absence of the analog (solid gray bar).

Moreover, in the groups treated with estradiol and antiestrogens, when EB1089 was added to these combinations either alone (E + T and E + F, striped black bars) or with lapatinib (E + T and E + F, striped gray bars), improved cell proliferation inhibition was achieved. The combined treatments involving EB1089, estradiol, and tamoxifen (E + T, striped black bar), as well as EB1089, lapatinib, estradiol, and fulvestrant (E + F, striped gray bar), significantly decreased cell proliferation compared to treatments that included only estradiol plus tamoxifen (E + T, black bar), or lapatinib, estradiol, and fulvestrant (E + F, solid gray bar), respectively.

Triple treatment of EB1089, lapatinib, and the antiestrogens in the absence of estradiol (T and F, striped gray bars) produced the highest inhibition of cell proliferation. A remarkable improvement in the inhibitory response was observed in the combination that included fulvestrant (F, striped gray bar). This effect differed significantly from each treatment alone (F, black, solid gray, and striped black bars).

These results indicate that adding EB1089 in the combined treatment of lapatinib with antiestrogens enhances the effectiveness of therapies targeting ER and HER2 in HER2-positive breast cancer cells expressing ERα.

Figure 2b shows that treatments with estradiol, tamoxifen, and fulvestrant (E, T, and F, black bars), as well as the combination of the antiestrogens with estradiol (E + T, E + F, black bars), did not have any effect on the proliferation of SK-BR-3 cells compared to the vehicle-treated cells (C, black bar). These results were expected because SK-BR-3 cells lack a therapeutic target for antiestrogens.

The treatments involving lapatinib, EB1089, and its combination (Figure 2b, group C, solid gray, striped black, and striped gray bars) demonstrated significant inhibition of cell proliferation compared to untreated cells (C, black bar). Likewise, this same pattern was observed when these treatments were co-administered with estradiol (E, solid gray, striped black, and striped gray bars), the antiestrogens (T and F, solid gray, striped black, and striped gray bars), and their combinations (E + T and E + F, solid gray, striped black, and striped gray bars) compared with their respective control (E, T, F, E + T, E + F, black bars).

In particular, within each group, the addition of EB1089 and lapatinib to treatments (C, E, E + T, E + F, T, and F, striped gray bars) showed a significant reduction in cell growth when compared to lapatinib treatments (C, E, E + T, E + F, T, and F, solid gray bars).

Remarkably, the combined treatment of EB1089, lapatinib, estradiol, and fulvestrant (E + F, striped gray bar) significantly reduced cell proliferation compared to each treatment separately (E + F, black, solid gray, and striped black bars). Moreover, triple treatment of EB1089, lapatinib, and fulvestrant (F, striped gray bar) produced the highest proliferation inhibition.

These findings confirm that EB1089 restores the antiproliferative response of antiestrogens and enhances the inhibitory effects of the combination of lapatinib with antiestrogens in HER2-positive breast cancer cells that lack ERα.

In addition, we conducted experiments involving dose–response curves for tamoxifen and fulvestrant in the presence of a constant concentration of EB1089 (EB, 1 × 10^−9^ M) and two concentrations of lapatinib (L-8, 1 × 10^−8^ M, and L-7, 1 × 10^−7^ M). For this, we selected BT-474 cells due to their expression of ERα. The results indicated that the addition of EB1089 and lapatinib at 1 × 10^−8^ M significantly reduced cell proliferation across all tamoxifen concentrations tested, compared to treatments without or with lapatinib at 1 × 10^−8^ M (Figure 3a).

Moreover, treatments comprising lapatinib at a concentration of 1 × 10^−7^ M combined with tamoxifen were significantly different from those without lapatinib. Furthermore, the combined treatment of EB1089 and lapatinib at 1 × 10^−7^ M exhibited the most cell growth inhibition across various tamoxifen concentrations. The treatment of EB1089, lapatinib at 1 × 10^−7^ M, and tamoxifen at 1 × 10^−7^ M and 1 × 10^−5^ M significantly decreased proliferation compared to the same treatments in the absence of analog.

The combination of EB1089 and lapatinib at 1 × 10^−8^ M significantly decreased cell growth across all fulvestrant concentrations, compared to treatments without or with lapatinib at 1 × 10^−8^ M (Figure 3b). Furthermore, the combination treatments involving lapatinib at the two concentrations with fulvestrant exhibited significant differences compared to treatments without lapatinib, except for the combination of lapatinib at 1 × 10^−8^ M and fulvestrant at 1 × 10^−9^ M.

Similar to the effects observed with tamoxifen treatments, the combination of EB1089 and lapatinib at a concentration of 1 × 10^−7^ M demonstrated the most inhibition of cell growth across various fulvestrant concentrations. These combinations were statistically significant compared to treatments without the analog.

The values of the IC_50_ were determined from the concentration–response curve of the combined treatments and are presented in Table 2.

The concentrations of antiestrogens alone or combined with lapatinib at 1 × 10^−8^ M did not exhibit a concentration-response curve, and they did not align with the function of the scientific analysis software. Interestingly, all values from the combined treatments involving tamoxifen demonstrated a similar IC_50_, indicating that the addition of EB1089 to the combinations of lapatinib at 1 × 10^−8^ M and tamoxifen yielded the same efficacy in inhibiting cell proliferation as the combinations of lapatinib at 1 × 10^−7^ M and tamoxifen. However, the addition of the analog did not alter the sensitivity of the cells to the combinations of lapatinib at 1 × 10^−7^ M and tamoxifen.

In contrast, the addition of EB1089 to the treatments of lapatinib at 1 × 10^−7^ M and fulvestrant resulted in a reduction of IC_50_ values by approximately 100 times compared to the absence of EB1089.

These results highlight a significant reduction in cell proliferation across all concentrations of antiestrogens tested upon the addition of EB1089 and lapatinib, compared to treatments without the analog.

### 2.3. The Addition of EB1089 to the Combination of Lapatinib and Antiestrogens Differentially Regulates ERα Protein Expression in HER2-Positive Breast Cancer Cells Depending on Cell Hormone Receptor Status

In breast cancer, the presence of ER serves as a favorable prognostic marker, as it is associated with less aggressive tumors, leading to higher overall survival rates and longer disease-free survival compared to ER-negative tumors [25]. Given the importance of ER in breast cancer and considering the differential effect of EB1089 on ERα expression in ER-positive and ER-negative breast cancer cells (as depicted in Figure 1), we investigated the effect of treatments on ERα protein expression in BT-474 and SK-BR-3 cells.

In the control group (C), treatment of BT-474 cells with tamoxifen (*p* ≤ 0.001), lapatinib, and their combination (*p* ≤ 0.001) (Figure 4a, T, L, L + T; and Figure 4c, T, black, C, solid gray, and T, solid gray bars) increased ERα protein expression when compared to cells treated with the compounds’ vehicles (Figure 4a, ethanol (V) or DMSO (D), for antiestrogens or lapatinib, respectively; and Figure 4c, C and D, black bars, respectively). Conversely, treatment with EB1089 (*p* ≤ 0.001, *t*-test) (Figure 4a,b, EB1089/EB; and Figure 4c, C, striped black bar) down-regulated ERα expression compared to the vehicle-treated cells (Figure 4a,b, C/V; and Figure 4c, C, black bar), consistent with our previous findings (Figure 1b, 1 × 10^−9^ M) and previous reports [19]. Furthermore, treatment with EB1089 combined with tamoxifen (*p* ≤ 0.001), lapatinib (*p* ≤ 0.05), or their combination (*p* ≤ 0.001) (Figure 4a, EB1089/T, L, L + T; and Figure 4c, T, striped black, C, striped gray, and T, striped gray bars) reversed the upregulation of ERα induced by these compounds alone or combined (Figure 4a, C/T, L, L + T; and Figure 4c, T, black, C, solid gray, and T, solid gray bars).

In contrast to tamoxifen, we observed the opposite effect with fulvestrant treatment (*p* ≤ 0.001, *t*-test) (Figure 4b, C/F; and Figure 4c, F, black bar) since it reduced ERα protein expression compared to untreated cells (Figure 4b, C/V and Figure 4c, C, black bar). Moreover, when lapatinib was combined with fulvestrant (*p* ≤ 0.01) (Figure 4b, C/L + F; and Figure 4c, F, solid gray bar) reversed the lapatinib-induced increase in ERα expression (Figure 4b, C/L; and Figure 4c, C, solid gray bar). Furthermore, the addition of EB1089 to fulvestrant (Figure 4b, EB1089/F; and Figure 4c, F, striped black bar) or the combination of lapatinib with fulvestrant (Figure 4b, EB1089/L + F; and Figure 4c, F, striped gray bar) induced a more pronounced ERα inhibition compared to the treatments without the analog (Figure 4b, C/F, C/L + F; and Figure 4c, F, black, F, solid gray bars).

These results indicate that the addition of EB1089 to treatments with lapatinib and antiestrogens and their combinations decreases ERα protein expression in HER2-positive breast cancer cells expressing ER. Notably, the EB1089-inhibitory effect in ER expression was more pronounced in all treatments involving fulvestrant than those with tamoxifen.

As previously demonstrated, calcitriol can induce ER protein expression on ER-negative breast cancer cells [12,13]. Accordingly, here, treatment of SK-BR-3 cells with EB1089 (*p* ≤ 0.01, *t*-test) (Figure 4d,e, EB1089/EB; and Figure 4f, C, striped black bar) upregulated ERα expression compared to the vehicle-treated cells (Figure 4d,e, C/V; and Figure 4f, C, black bar), in agreement with the results shown in Figure 1c. Tamoxifen (*p* ≤ 0.001, *t*-test), fulvestrant (*p* ≤ 0.005, *t*-test), and lapatinib alone (*p* ≤ 0.01, *t*-test) (Figure 4d,e, C/T, C/F, C/L; and Figure 4f, T, black, F, black and C, solid gray bars), as well as their combinations (*p* ≤ 0.01, *t*-test) (Figure 4d,e, C/L + T, C/L+ F; and Figure 4f, T, solid gray, F, solid gray bars), also upregulated the expression of ERα. The addition of EB1089 to these treatments did not modify this effect (Figure 4d,e, EB1089/T, EB1089/L, and EB1089/L+ F; and Figure 4f, T, striped black, C, striped gray, F, striped gray bars), except in the treatments with fulvestrant and in the combination of lapatinib and tamoxifen (Figure 4c, EB1089/F and EB1089/L + T; and Figure 4f, F, striped black, T, striped gray bars) that showed tend to reduce the ERα-induced.

These results indicate that the treatments with EB1089, lapatinib, and antiestrogens increase ERα protein expression in HER2-positive breast cancer cells that lack ERα.

### 2.4. The Addition of EB1089 to Lapatinib and Antiestrogens Inhibits Akt Phosphorylation in BT-474 Breast Cancer Cells

Activation of the PI3K/AKT/mTOR signaling pathway, whether due to mutations in pathway components or activation of upstream signaling molecules, is common in HER2-positive breast cancer. This activation contributes to the dysregulation of cell proliferation, apoptosis resistance, and metabolism changes [26,27].

Considering that the best anti-proliferative response was observed in the combined treatment of EB1089, lapatinib, and fulvestrant, we decided to evaluate the effect of this combination on the activation of crucial proteins in HER2-activated downstream pathways in BT-474 cells, using an Akt microarray kit. Analysis of the results, as depicted in Figure 5a, revealed that among the 18 proteins tested using the microarray kit, only eight were detected, of which seven underwent reduction of their phosphorylated form expression (pAKT, pBAD, p4E-BP1, pGSK3B, pMTOR, pP70S6K, pPRAS40) in response to the triple treatment (Figure 5b). Notably, the triple treatment (Figure 5b, AKT gray bar) induced a considerable reduction in Akt phosphorylation compared to the control (Figure 5b, AKT black bar).

Based on these findings, we performed Western blot analyses to evaluate the effect of the treatments on the phosphorylation status of Akt and Src. This last protein is crucial in regulating cell cycle progression and mediating cell survival by activating the PI3K/AKT pathway [28].

The results confirmed the inhibition of Akt phosphorylation by the triple treatment regimen with fulvestrant (Figure 5d, EB1089/L + F; and Figure 5e, F, striped gray bar). In addition, all treatment conditions, including individual treatments or combined, inhibited Akt phosphorylation compared to the vehicle-treated cells (Figure 5c,d, C/V; and Figure 5e, C, black bar). Importantly, combinations of EB1089 with the antiestrogens (Figure 5c,d, EB1089/T and EB1089/F; and Figure 5e, T, striped black, F, striped black bars), lapatinib (Figure 5c,d, EB1089/L; and Figure 5e, C, striped gray bar), or their combinations (Figure 5c,d, EB1089/L + T and EB1089/L + F; and Figure 5e, T, striped gray, F, striped gray bars), exhibited more pronounced inhibition of Akt phosphorylation compared to the treatments in the absence of the analog (Figure 5c,d, C/T, C/F, C/L, C/L + T, and C/L + F; and Figure 5e, T, black, F, black, C, solid gray, T, solid gray, F, solid gray bars).

On the other hand, the antiestrogens or lapatinib by themselves or the combined treatment of the TKI with tamoxifen (Figure 5f,g, C/T, C/F, C/L, and C/L + T; and Figure 5h, T, black, F, black, C, solid gray, T, solid gray bars) increased Src phosphorylation compared to non-treated cells (Figure 5f,g, C/V; and Figure 5h, C, black bar). In contrast, EB1089 (Figure 5f,g, EB1089/EB; and Figure 5h, C, striped black bar) inhibited Src phosphorylation, and the addition of the analog to these treatments (Figure 5f,g, EB1089/T and EB1089/F, EB1089/L, EB1089/L + T; and Figure 5h, T, striped black, F, striped black, C, striped gray, T, striped gray bars) reverted the increment on Src phosphorylation.

Notably, the combination of lapatinib and fulvestrant (Figure 5g, C/L + F; and Figure 5h, F, solid gray bar) showed a substantial reduction in Src phosphorylation, an effect that was reverted by the addition of the analog (Figure 5g, EB1089/L + F; and Figure 5h, F, striped gray bar). However, this combination’s inhibition of Src phosphorylation remained below that of untreated cells (Figure 5g, C/V; and Figure 5h, C, black bar).

These data indicate that adding EB1089 in the combination treatment of lapatinib and antiestrogens enhances the inhibition of Src/Akt phosphorylation in HER2-positive breast cancer cells that express ERα.

### 2.5. The Addition of EB1089 to Lapatinib and Antiestrogens Inhibits Akt Phosphorylation in the SK-BR-3 Breast Cancer Cells

We also assessed the effect of the treatments on the phosphorylation status of Akt and Src in the SK-BR-3 cells.

Overall, treatment in the absence of the analog (Figure 6a,b, C/T, C/F, C/L, C/L + T, C/L + F; Figure 6c, T, black, F, black, C, solid gray, T, solid gray, F, solid gray bars) showed a slight reduction of Akt phosphorylation compared to non-treated cells (Figure 6a,b, C/V; Figure 6c, C, black bar). Notably, EB1089 per se (Figure 6a,b, EB1089/EB; Figure 6c, C, striped black bar) as well as its addition to treatments alone (Figure 6a,b, EB1089/T and EB1089/F, EB1089/L; Figure 6c, T, striped black, F, striped black, C, striped gray bars) or combined (Figure 6a,b, EB1089/L + T, EB1089/L + F; Figure 6c, T, striped gray, F, striped gray bars) increased the inhibition of Akt phosphorylation compared to the treatments without the analog, except in the lapatinib and fulvestrant treatment (Figure 6b, C/L + F), the addition of EB1089 did not alter the phosphorylated form of Akt (Figure 6b, EB1089/L + F; Figure 6c, F, striped gray bar).

Densitometric analysis showed that tamoxifen, lapatinib alone or combined with antiestrogens treatments (Figure 6d,e, C/T, C/L, C/L + T, C/L + F; Figure 6f, T, black, C, solid gray T, solid gray, F, solid gray bars) slightly reduced Src phosphorylation compared to non-treated cells (Figure 6d,e, C/V; Figure 6f, C, black bar).

Overall, fulvestrant treatment (Figure 6e, C/F; Figure 6f, F, black bar) did not affect Src phosphorylation compared to the cells treated with the vehicle (Figure 6e, C/V; Figure 6f, C, black bar) or when EB1089 was added to antiestrogens, lapatinib, or their combinations.

These data indicate that adding EB1089 in the combination treatment of lapatinib and tamoxifen enhances the inhibition of Akt phosphorylation in breast cancer cells that overexpress HER2 and lack ERα.

## 3. Discussion

Despite the design of more specific drugs targeting molecules and pathways involved in breast cancer, drug resistance remains a significant concern in cancer therapy. The dual tyrosine kinase inhibitor lapatinib, which targets EGFR and HER2, has been approved by the FDA for HER2-positive breast cancer [29]. However, patients receiving lapatinib often develop resistance, and one of the major mechanisms responsible for this resistance is the activation of the ER pathway [23,30]. Therefore, combining lapatinib with therapies that block ER signaling has been explored to prevent the development of lapatinib resistance [30,31].

The most active metabolite of vitamin D, calcitriol, has demonstrated antineoplastic properties against various cancers, including breast cancer [32,33,34]. However, the issue of hypercalcemia induced by high-dose vitamin D has led to the synthesis of analogs with reduced hypercalcemic effects. One such analog is EB1089, which exhibits less toxicity and more potent inhibitory effects on cell growth than calcitriol [14,16,35]. Combining EB1089 with other therapies, including endocrine and anti-HER2 treatments, has shown promising outcomes in overcoming resistance and enhancing treatment efficacy in breast cancer, both in in vitro and in vivo studies [9,11,24,36,37]. A previous study has shown synergetic effects between lapatinib and antiestrogens in breast cancer [31]. Additionally, combinations of EB1089 with antiestrogens have demonstrated synergistic and additive effects [36]. However, the combination of EB1089 with antiestrogens and lapatinib has not been explored. Therefore, in this study, we aimed to evaluate the effects of adding EB1089 to the combined treatment with lapatinib and antiestrogens in HER2-positive breast cancer cells using two cell lines representing distinct HER2-positive breast cancer phenotypes: BT-474 that is characterized by its expression of ER (ER-positive/HER2-positive) and SK-BR-3 that lacks ER (ER-negative/HER2-positive).

Modulation of ER expression was shown to be part of the mechanism of EB1089′s growth inhibition capacity, resulting in reduced responsiveness of cells to the growth-stimulatory effects of estradiol [20]. Previous studies found that while antiestrogens alone caused inhibition of estradiol-stimulated growth, adding calcitriol or EB1089 further enhanced the growth inhibition of estradiol-stimulated ER-positive cells [20,24,38]. Consistent with these findings, our results with BT-474 cells demonstrated that the addition of EB1089 to estradiol alone or in combination with the antiestrogens reduced estradiol-stimulated proliferation, confirming that the addition of EB1089 to endocrine treatment enhances the antiproliferative effects of the antiestrogens, even in the presence of estradiol. Likewise, BT-474 cells treated with combinations of lapatinib with anti-estrogens or EB1089 with lapatinib significantly inhibited cell growth, which aligns with previous research findings [9,31,39,40].

The study’s contribution lies in the results obtained from the triple compounds treatment, including EB1089, lapatinib, and the antiestrogens in BT-474 cells, which showed the highest inhibition of proliferation among all treatments. Of note, the combination involving fulvestrant showed the most potent effect. These results suggest that the addition of EB1089 to the treatment regimen of lapatinib and antiestrogens could help overcome resistance to HER2- and ER-targeted therapies while enhancing the inhibitory response of these compounds. This combination approach offers a promising strategy for improving outcomes in patients with ER-positive and HER2-positive breast cancer.

Previous studies from our laboratory and other research groups have demonstrated the antiproliferative effects of calcitriol and its analog EB1089 in ER-negative breast cancer cell lines [20,24,41,42], consistent with the findings presented in this study. We previously demonstrated that calcitriol restored antiestrogens antiproliferative effects in ER-negative breast tumor-derived cells and the triple-negative breast cancer cell line SUM-229PE [12]. Similarly, our current results showed that EB1089 treatment inhibited cell proliferation and restored the antiestrogenic responsiveness of fulvestrant in SK-BR-3 cells, indicating the effectiveness of this combination therapy in inhibiting proliferation in breast cancer cells with the ER-negative/HER2-positive phenotype. Notably, no proliferative activity was observed when EB1089 was combined with estradiol, suggesting that the analog counteracts the agonist’s proliferative effects. This observation aligns with our previous findings, demonstrating that estradiol did not impact cell growth in calcitriol-treated ER-negative tumor-derived cells [12].

In our investigation, we focused on examining the impact of different concentrations of EB1089 (1 × 10^−9^ M, 1 × 10^−8^ M, and 1 × 10^−7^ M) when combined with antiestrogens on the antiproliferative response. Notably, 1 × 10^−9^ M concentration enhanced the antiproliferative effects of the combined treatment in both BT-474 and SK-BR-3 HER2-positive breast cancer cells. These findings underscore the potential of utilizing a low concentration of EB1089 (1 × 10^−9^ M) to augment the antiproliferative effects of the treatments significantly.

Moreover, experiments involving dose–response curves for antiestrogens in the presence of EB1089 and lapatinib in BT-474 cells demonstrated a significant reduction in cell proliferation across all tested concentrations of antiestrogens upon the addition of EB1089 and lapatinib. This contrasted with treatments without the analog, indicating sensitivity changes across various concentrations. Ultimately, this contributes to the potential for reducing the concentration of antineoplastics while enhancing their antiproliferative effects.

Therapies targeting HER2 often encounter resistance through various escape mechanisms, with the involvement of the ER pathway being a particularly common widespread [23,43,44]. Specifically, the dynamic interaction between HER2 and ER activity plays a crucial role in determining resistance to lapatinib-based treatments [23,30]. Therefore, a combined approach of administering endocrine and anti-HER2 therapies could benefit patients with ER-positive/HER2-positive, even in cases where tumors are clinically classified as ER-negative [23]. Our study’s results support this approach by showing that combining lapatinib with fulvestrant improves growth inhibition than each treatment alone in SK-BR-3 cells.

Consistent with our observations in BT-474 cells, triple treatment involving lapatinib, the antiestrogens, and EB1089 demonstrated the most potent antiproliferative effects in SK-BR-3 cells. These results provide further support for our hypothesis, highlighting the potential of a combined therapeutic approach integrating HER2-targeted therapy, endocrine therapy, and the antiproliferative properties of EB1089. This approach holds promise in overcoming resistance pathways often associated with monotherapy or dual therapy in patients with ER-negative/HER2-positive breast cancer. To expand the scope and depth of our investigations, further studies should encompass a more diverse range of cell lines, including drug-resistant derivatives from each cell line, and consider incorporating in vivo models. These steps will enable a more comprehensive understanding of the therapeutic potential and validate the translational relevance of our findings for clinical applications.

ER is a favorable prognostic marker in breast cancer, and its presence is associated with less aggressive tumors, leading to higher overall survival rates and longer disease-free survival than ER-negative tumors [25]. Patients with ER-negative/HER2-positive breast cancer experience earlier relapses than those with ER-positive/HER2-positive disease [45]. Patients with ER-positive tumors also tend to respond favorably to endocrine therapy [46,47,48]. Our results demonstrated that EB1089 decreases ERα protein expression in BT-474 cells and induces receptor expression in SK-BR-3 cells. These findings align with previous findings, where calcitriol modulated ER protein expression depending on the receptor status [12,13,20]. These results suggest that EB1089 affects the phenotype of HER2-positive breast cancer cells by modulating ERα expression, reversing cellular mechanisms associated with more aggressive behavior and poor prognosis.

In this study, we observed that BT-474 cells treated with fulvestrant decreased ERα protein expression. In contrast, tamoxifen treatment produced the opposite effect, which aligns with previous research findings [49,50,51,52]. Like tamoxifen, lapatinib upregulated the expression of ERα in BT-474 cells, a phenomenon observed previously [30]. The lapatinib-induced upregulation of ERα has been attributed to the compensatory activation of the ERα pathway following the inhibition of HER2, AKT, and ERK phosphorylation by lapatinib. This suggests that the compensatory activation of ER signaling may contribute to acquired resistance to lapatinib in HER2-positive/ERα-positive breast cancer [30]. Conversely, we observed that combining lapatinib with fulvestrant reversed the lapatinib-induced increase of ERα expression, which aligns with previous findings [30,53]. Notably, the addition of EB1089 to tamoxifen and lapatinib, or their combination, reversed the upregulation of ERα induced by these compounds. Furthermore, EB1089, lapatinib, and fulvestrant combination showed the most pronounced inhibitory effect on ER expression. These results suggest that triple treatment combining EB1089 with lapatinib and antiestrogens could help overcome acquired resistance caused by every single treatment through ERα expression regulation.

Similarly to EB1089, the treatment of SK-BR-3 cells with lapatinib, antiestrogens, and their combinations significantly upregulated the expression of ERα. Concerning this, in vitro studies have shown that early exposure to tamoxifen led to modifications in an ERα-dependent subset of genes associated with developmental processes and pluripotency in ER-negative MDA-MB-231 cells [54]. Furthermore, treatment with gefitinib, a tyrosine kinase inhibitor, resulted in the re-expression of ERα in ERα-negative breast cancer cell lines like SUM 190 and SUM-229PE by inhibiting the MAPK pathway [12,55]. This aligns with our findings, where lapatinib treatment induced ERα expression in SK-BR-3 cells. However, in certain cell lines, such as SUM 102 and SUM 159, characterized by a basal phenotype and hypermethylation of the ERα promoter, it was observed that MAPK inhibition alone did not restore ERα expression [55]. Indeed, calcitriol has been shown to induce ERα expression through direct transcriptional regulation and epigenetic modifications in estrogen receptor-negative breast cancer cells [13]. These findings suggest that additional mechanisms, such as demethylation of the ERα promoter or regulation of epigenetic mechanisms, may also play a role in the induction of ERα expression. Further studies are necessary to fully understand the potential implications of ERα re-expression regarding prognosis and strategies to overcome therapy resistance in HER2-positive breast cancer.

The dysregulation of the PI3K/AKT/mTOR pathway is associated with tumor progression and resistance to HER2-targeted therapies in breast cancer [56,57,58,59,60]. Therefore, we evaluated the effects of the treatments on signal molecules downstream of these pathways in HER2-positive breast cancer cells.

In line with calcitriol’s effects on breast cancer cells and tumors [61,62], EB1089 decreased phosphorylated Akt expression. Likewise, the antiestrogens and lapatinib inhibited AKT phosphorylation, as reported by other authors [22,63,64]. Interestingly, the addition of the vitamin D analog to lapatinib, the antiestrogens alone or in combination, exhibited a notable inhibition of Akt phosphorylation in BT-474 cells.

The combined treatment of EB1089, lapatinib, and fulvestrant suppressed, besides AKT/mTOR pathway activity, the downstream effectors 4EBP1 (eukaryotic translation initiation factor 4E binding protein 1), P70S6K (ribosomal protein S6 kinase B1), and PRAS40 (AKT1 substrate 1), which play crucial roles in regulating translation [65,66,67,68,69]. Additionally, this treatment inhibited the phosphorylation of BAD (Bcl-2 antagonist of cell death BAD), a key mediator of the apoptotic pathway that plays a significant role in carcinogenesis and chemo-response [70,71], as well as GSK3β, which inhibits apoptosis, is highly expressed in breast cancer tissues, and has been associated with poor survival outcomes [72,73,74]. Our findings suggest that the triple combination of EB1089, lapatinib, and antiestrogens inhibits proliferation by reducing AKT/mTOR pathway activity, subsequently affecting protein synthesis, apoptosis, and cell growth.

Src, a non-receptor tyrosine kinase, is associated with developing resistance against HER2-targeted therapy in breast cancer cells [75,76]. Additionally, it plays a critical role in mediating tamoxifen resistance, and blocking its activity has been shown to reverse tamoxifen resistance [77]. Interestingly, EB1089 considerably inhibited Src phosphorylation, and when combined with antiestrogens or lapatinib and their combinations, it produced a similar effect in the BT-474 cells. These results suggest that the inhibition of Src phosphorylation may be a mechanism through which these treatments can overcome resistance to therapies.

In SK-BR-3 cells, the addition of EB1089 to the antiestrogens and the combination of EB1089, lapatinib, and tamoxifen resulted in a more potent reduction of Akt phosphorylation. However, when EB1089 was added to the combination of lapatinib and fulvestrant, it did not further enhance Akt phosphorylation inhibition as observed with the combined treatment of EB1089, lapatinib, and tamoxifen. This may be attributed to the increased activation of Akt that is commonly observed in fulvestrant-resistant breast cancer cell lines [78,79]. Overall, the treatments did not notably affect Src phosphorylation in this particular cell line. Several factors might contribute to this lack of effect, including the concentrations utilized in this experiment. Moreover, the distinct ER/HER2 statuses between BT-474 (ER-positive/HER2-positive) and SK-BR-3 (ER-negative/HER2-positive) cells, alongside potential variations in cell differentiation, could be influential factors contributing to these differences. These variations likely reflect the diverse phenotypes and varying levels of aggressiveness among different cancer cell types. These results suggest that the inhibition of Akt phosphorylation may be a mechanism through which EB1089, lapatinib, and tamoxifen treatment inhibit cell proliferation.

## 4. Materials and Methods

### 4.1. Reagents

Cell culture media were obtained from Life Technologies (Grand Island, NY, USA). Fetal Bovine Serum (FBS) was purchased from Hyclone Laboratories Inc. (Logan, UT, USA). Lapatinib was purchased from Sequoia Research Products (Pangbourne UK). EB1089 (seocalcitol) was obtained from Tocris Bioscience (Bristol, UK). Estradiol (E2) and 4-Hydroxytamoxifen were purchased from Sigma (St. Louis, MO, USA), and antiestrogen fulvestrant (ICI-182780) from Zeneca Pharmaceuticals (Wilmington, DE, USA). Human and Mouse AKT Pathway Phosphorylation Array C1 protein micro-array kit was purchased from Ray Biotech (GA, USA).

### 4.2. Cell Lines and Culture

Two HER2-positive breast cancer cell lines characterized by the expression of ER, BT-474 (ER-positive/HER2-positive), or its absence, SK-BR-3 (ER-negative/HER2-positive), were used in this study [80]. The cell lines were obtained from the American Type Culture Collection (ATCC; Rockville, MD, USA). Cell lines were cultured in Hybri-Care (BT-474) and McCoy’s 5A (SK-BR-3) media supplemented with 10% of fetal bovine serum (FBS) and 1% of Penicillin (100 U/mL)/Streptomycin (100 μg/mL). All experimental procedures were performed in Dulbecco’s modified Eagle’s medium (DMEM) supplemented with 10% charcoal-stripped-heat-inactivated FBS, 100 U/mL penicillin, and 100 µg/mL streptomycin.

### 4.3. Treatments

To modulate ERα expression, BT-474 or SK-BR-3 cells were pre-treated with EB1089 (1 × 10^−9^ M) or its vehicle (ethanol 0.01%) as a control for 48 h. Subsequently, the cells were treated with lapatinib (1 × 10^−8^ M or 1 × 10^−7^ M), tamoxifen (1 × 10^−9^ M, 1 × 10^−7^ M, 1 × 10^−6^ M or 1 × 10^−5^ M) or fulvestrant (1 × 10^−9^ M, 1 × 10^−7^ M, 1 × 10^−6^ M or 1 × 10^−5^ M), either alone or in combination with estradiol (1 × 10^−8^ M), in the absence or presence of lapatinib. All treatments were prepared using the corresponding pre-treatment media. The treatment durations for gene expression, protein level, and cell proliferation analyses were 24 h, 48 h, and 72 h, respectively.

### 4.4. Cell Proliferation Assay

Cells were seeded in 96-well plates (2000 cells/well) for 24 h. Subsequently, the cells were pre-treated and treated as described earlier. Cell proliferation was evaluated by the sulforhodamine B (SRB) colorimetric assay, as previously described [13]. Briefly, the cells were fixed with 10% trichloroacetic acid and incubated at 4 °C for 1 h. The plates were then gently washed with tap water and air-dried. Next, the cells were stained with 0.4% SRB dissolved in 1% acetic acid for 1 h at room temperature. To remove the unbound stain, the plates were washed four times with 1% acetic acid and air-dried. The bound protein stain was solubilized with 10 mM of unbuffered Tris base [tris(hydroxymethyl) aminomethane]. Absorbance was measured at 492 nm using a microplate reader (Synergy HT Multi-Mode Microplate Reader BioTek, Winooski, VT, USA). The values of the IC_20_ and IC_50_ of EB1089 were determined from the dose–response curve and the dose–response fitting function of the scientific graphing and analysis software OriginPro 8.lnk (OriginLab Corporation, Northampton, MA, USA, version 8.0).

### 4.5. Western Blot

BT-474 or SK-BR-3 cells were seeded in Petri dishes and treated as previously described. Whole-cell protein lysates were then prepared using a lysis buffer supplemented with protease and phosphatase inhibitors (PPC1010, Sigma). Protein concentrations were determined using the Bradford method (Bio-Rad, Hercules, CA, USA). Proteins were separated on 10% SDS-PAGE and transferred to PVDF membranes. The membranes were then blocked with 5% skim milk for 1 h at room temperature and incubated with the following primary antibodies: ERα (F10, sc-8002, Santa Cruz Biotechnology, CA, USA), phospho-Akt (Ser473, #9271, Cell Signaling Technology, MA, USA), total Akt (#9272, Cell Signaling Technology, MA, USA), phospho-Src (H-3, sc166860, Santa Cruz Biotechnology, CA, USA), and total Src (B-12, sc-8056, Santa Cruz Biotechnology, CA, USA), overnight at 4 °C. The membranes were washed and incubated with goat anti-rabbit (sc-2004, Santa Cruz Biotechnology, CA, USA) or goat anti-mouse (sc-2055, Santa Cruz Biotechnology, CA, USA) HRP-conjugated secondary antibodies. For visualization, the membranes were processed with BM chemiluminescence blotting substrate (Roche Applied Science MA, DE). For normalization, the antibodies present on the membrane were removed with stripping buffer (Glycine; Mg Bicarbonate (CH_3_COO) 2-4H_2_O; KCl; pH 2.2, Sigma-Aldrich products, MO, USA) for 20 min, and subsequently, the membranes were incubated with GAPDH (6C5, sc-32233, Santa Cruz Biotechnology, CA, USA) and then goat anti-mouse-HRP (AB_2313585, Jackson ImmunoResearch Laboratories, Inc., West Grove, PA, USA). Densitometric analysis of the bands was performed using the ImageJ software version 1.52a (NIH, Bethesda, MD, USA).

### 4.6. Microarray Assay

The RayBiotech C-Series Human and Mouse AKT Pathway Phosphorylation Array C1 kit was employed for this assay. BT-474 cells were seeded in Petri dishes (2 × 10^5^ cells) and divided into control and treated groups. Both groups underwent pre-treatment as previously described. For the treatment, the control group was maintained in a freshly prepared medium under the same conditions as the pre-treatment. In contrast, the cells in the treated group were incubated in a medium containing EB1089 (1 × 10^−9^ M), lapatinib (1 × 10^−8^ M), and fulvestrant (1 × 10^−7^ M) for 48 h. At the end of the treatment period, the media were aspirated, and the lysis buffer provided in the kit was used for whole-cell extraction. The protein concentration was quantified using the Bradford method (Bio-Rad, Hercules, CA, USA).

The protein array assay was performed according to the manufacturer’s protocol with slight modifications. Briefly, the antibody membranes were blocked at room temperature for 30 min. Subsequently, the samples were added and incubated overnight at 4 °C. The detection antibody cocktail was prepared, pipetted onto the membranes, and incubated overnight at 4 °C. Finally, the samples were incubated with the provided HRP secondary antibody and visualized using the ChemiDoc Touch Imaging System (ChemiDoc, Bio-Rad, CA, USA) for the semi-quantitative detection of 18 phosphorylated human and mouse proteins: Akt (P-Ser473); PKB (P-Ser473); AMPKa (P-Thr172); BAD (P-Ser112); 4EBP1 (P-Thr36); ERK1 (P-T202/Y204)/ERK2 (P-Y185/Y187); GSK3a (P-Ser21); GSK3b (P-Ser9); mTOR (P-Ser2448); p27 (P-Thr198); P53 (P-Ser15); P70S6K (P-Thr421/Ser424); PDK1 (P-Ser241); PRAS40 (P-Thr246); PTEN (P-Ser380); Raf-1 (Ser301); RPS6 (P-Ser235/Ser236); RSK1 (P-Ser380); RSK2 (P-Ser386).

### 4.7. Statistical Analysis

Results are expressed as mean ± SD and statistical analyses were determined by one-way ANOVA followed by the Holm–Sidak method. Furthermore, in some experiments, the *t*-student test was used for statistical analysis. Differences were considered significant at *p* ≤ 0.001 and 0.05.

## 5. Conclusions

EB1089 enhances the antiproliferative activity of endocrine treatment alone or in combination with lapatinib in BT-474 (ER-positive/HER2-positive) breast cancer cells. Furthermore, EB1089 restores antiestrogen responsiveness and increases the antineoplastic activity of the combined lapatinib treatment with antiestrogens in SK-BR-3 (ER-negative/HER2-positive) breast cancer cells. These effects are mediated through the modulation of ERα expression and the inhibition of Akt phosphorylation. These findings suggest that EB1089 could be a promising therapeutic option to enhance treatment outcomes in HER2-positive breast cancer, especially in cases where antiestrogen therapy is less effective due to lack of ER expression.

## Figures and Tables

**Figure 1 ijms-25-03165-f001:**
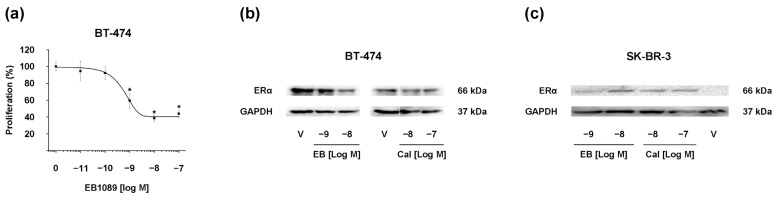
EB1089 modulates cell proliferation and ERα protein expression in HER2-positive breast cancer cells. (**a**) BT-474 cells were treated with either vehicle (ethanol, 0.01%) or increasing concentrations of EB1089 for three days. Subsequently, cell viability was assessed using the Sulforhodamine B assay. Results are expressed as mean ± S.D. of triplicate determinations and represent two experiments. Data from vehicle-treated cells were normalized to 100%. * *p* < 0.001 vs. vehicle. (**b**) BT-474 and (**c**) SK-BR-3 cells were treated with EB1089 (EB) at 1 × 10^−9^ M and 1 × 10^−8^ M, calcitriol (Cal) at 1 × 10^−8^ M and 1 × 10^−7^ M, or their vehicle ethanol (V). After 48 h, the media were replaced with fresh media containing the same compounds, and the cells were further incubated for 48 additional hours. A representative image from two independent experiments is shown.

**Figure 2 ijms-25-03165-f002:**
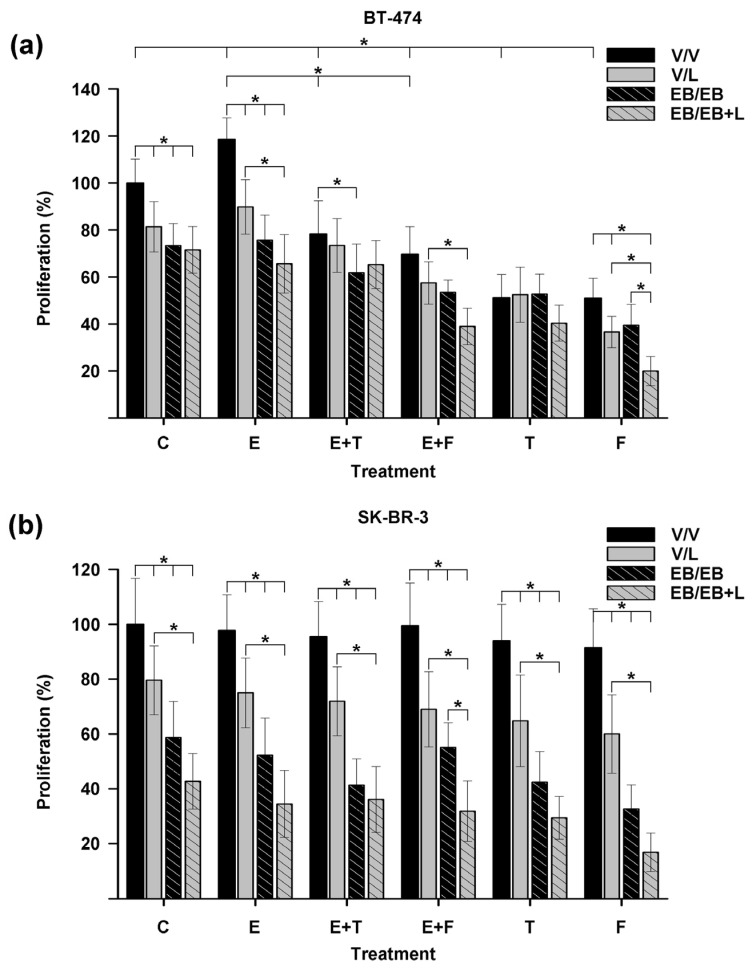
Effects of the addition of EB1089 to the combined treatment of lapatinib and antiestrogens on cell proliferation in HER2-positive breast cancer cells. (**a**) BT-474 or (**b**) SK-BR-3 cells were pre-treated with EB1089 (EB, 1 × 10^−9^ M) or its vehicle (V) for 48 h. Subsequently, the cells were treated for three days with lapatinib (L, 1 × 10^−8^ M), tamoxifen (T, 1 × 10^−6^ M), or fulvestrant (F, 1 × 10^−6^ M) alone or in combination with estradiol (E, 1 × 10^−8^ M). These treatments were performed in the absence or presence of the analog, and cell proliferation assays were performed. Results are expressed as mean ± S.D. of triplicate determinations and represent at least three independent experiments. Data were normalized to 100% using the values of vehicle-treated cells. * *p* < 0.05.

**Figure 3 ijms-25-03165-f003:**
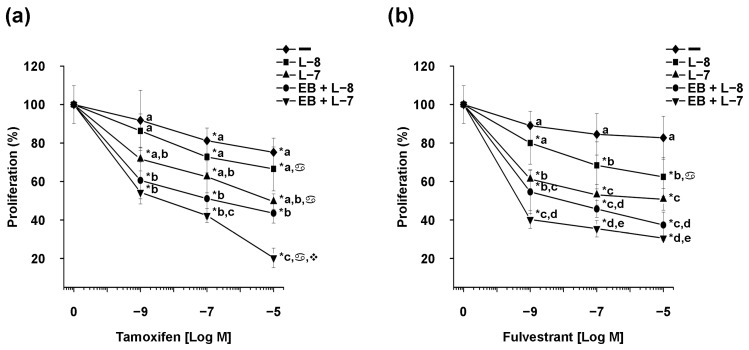
Effect of antiestrogens on HER2-positive breast cancer cell proliferation in the presence of EB1089 and lapatinib. BT-474 cells were pretreated with EB1089 (EB, 1 × 10^−9^ M) or its vehicle for 48 h. Subsequently, the cells were treated without (−) or with lapatinib (L, 1 × 10^−8^ M, and 1 × 10^−7^ M) combined with different (**a**) tamoxifen and (**b**) fulvestrant concentrations (1 × 10^−9^ M, 1 × 10^−7^ M, and 1 × 10^−5^ M) for six days. These treatments were done in the absence or presence of the analog, and cell proliferation assays were performed. Results are expressed as mean ± S.D. of sextuplicate determinations and represent two experiments. Data were normalized to 100% using the values of vehicle-treated cells. * *p* < 0.001 vs. non-treated cells (0), different letters *p* < 0.001 vs. same concentration of antiestrogen treatment, 
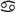

*p* < 0.001 vs. antiestrogen treatment 1 × 10^−9^ M, ❖ *p* < 0.001 vs. antiestrogen treatment 1 × 10^−7^ M.

**Figure 4 ijms-25-03165-f004:**
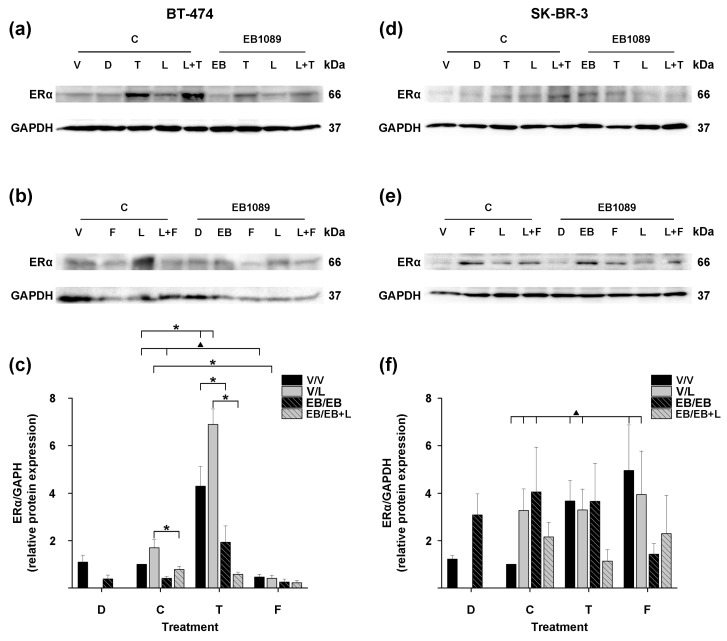
Modulation of ERα protein expression by the combination treatment of EB1089, lapatinib, and antiestrogens in HER2-positive breast cancer cells. (**a**–**c**) BT-474 and (**d**–**f**) SK-BR-3 cells were pretreated for 48 h in the presence of EB1089 (1 × 10^−9^ M) or its vehicle (V). Subsequently, the cells were treated for 48 h with tamoxifen (T, 1 × 10^−7^ M), fulvestrant (F, 1 × 10^−7^ M), lapatinib (L, 1 × 10^−8^ M) lapatinib vehicle DMSO (D), or the combinations of antiestrogens with the tyrosine kinase inhibitor. These treatments were performed in the absence or presence of EB1089. Western blot analysis detected ERα protein, and GAPDH was utilized as the loading control. (**a**,**d**) Representative blots of treatments involving tamoxifen or (**b**,**e**) fulvestrant from three independent experiments are shown. * *p*, one-way ANOVA. ^▲^
*p*, *t*-test.

**Figure 5 ijms-25-03165-f005:**
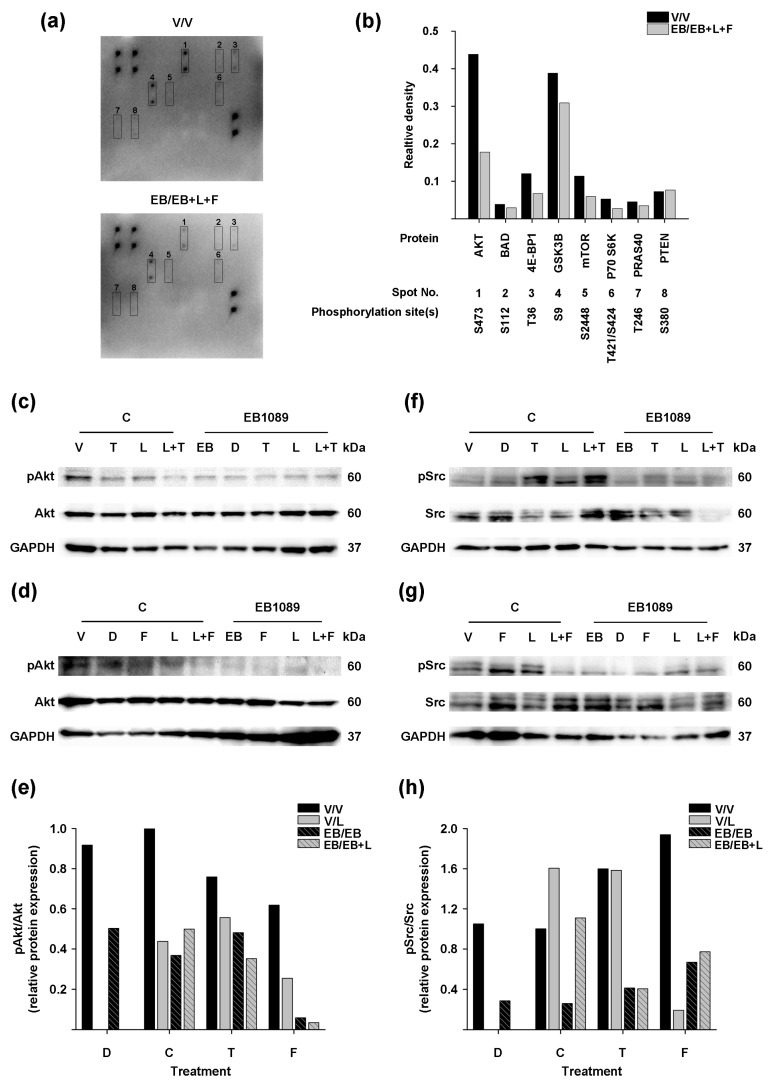
Effect of the combination of EB1089, lapatinib, and antiestrogens in Src-Akt signaling in BT-474 cells. (**a**) Representative arrays and (**b**) densitometric analysis of proteins crucial to the AKT pathway of BT-474 cell lysates. Cells were pre-treated in the presence of EB1089 (EB, 1 × 10^−9^ M) or its vehicle (V) for 48 h. Subsequently, the cells were treated for two days in the absence or presence of EB1089, lapatinib (L, 1 × 10^−8^ M), and fulvestrant (F, 1 × 10^−7^ M). In addition, BT-474 cells were treated as described in Figure 4. Blots detected Akt (**c**,**d**) and Src (**f**,**g**) total and its phosphorylated form, along with their respective densitometric analyses (**e**,**h**). GAPDH was used as the loading control. (**c**,**f**) Representative blots of treatments involving tamoxifen or (**d**,**g**) fulvestrant from two independent experiments are shown.

**Figure 6 ijms-25-03165-f006:**
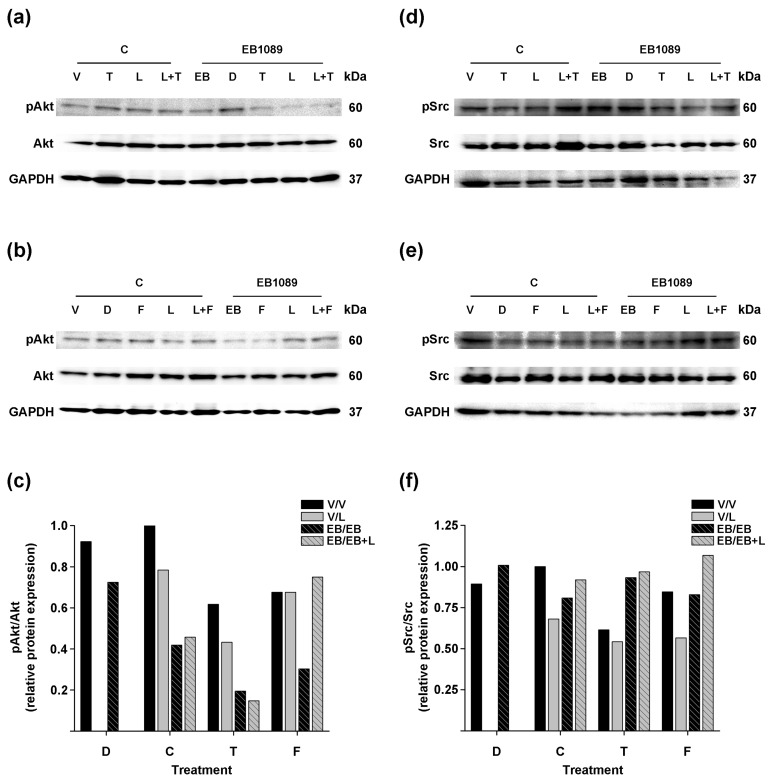
Effect of the combination of EB1089, lapatinib, and antiestrogens in Src-Akt signaling in SK-BR-3 cells. Cells were treated as described in Figure 4. Blots detected Akt (**a**,**b**) and Src (**d**,**e**) total and its phosphorylated form, along with their respective densitometric analyses (**c**,**f**). GAPDH was used as the loading control. (**a**,**d**) Representative blots of treatments involving tamoxifen or (**b**,**e**) fulvestrant from two independent experiments are shown.

**Table 2 ijms-25-03165-t002:** Inhibitory concentration values of the combined treatments of EB1089, lapatinib, and antiestrogens on the proliferation of BT-474 breast cancer cells.

Treatments		mol/L
**Tamoxifen +**	−	nd
	L − 8	nd
	EB + L − 8	6.67 × 10^−8^ ± 1.08 × 10^−7^
	L − 7	3.93 × 10^−8^ ± 8.09 × 10^−8^
	EB + L − 7	7.56 × 10^−8^ ± 8.46 × 10^−8^
**Fulvestrant +**	−	nd
	L − 8	nd
	EB + L − 8	3.48 × 10^−8^ ± 1.020 × 10^−7^
	L − 7	3.72 × 10^−8^ ± 4.40 × 10^−8^
	EB + L − 7	2.13 × 10^−10^ ± 5.40 × 10^−11^

Non-treatment (−), lapatinib 1 × 10^−8^ M (L − 8) and 1 × 10^−7^ M (L − 7), EB1089 1 × 10^−9^ M (EB), not determined (nd).

## Data Availability

The authors confirm that the data supporting the findings of this study are available within the article.

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
