# Peer review of "EB1089 Increases the Antiproliferative Response of Lapatinib in Combination with Antiestrogens in HER2-Positive Breast Cancer Cells"

_ijms, 2024, doi:10.3390/ijms25063165_

Round 1
Reviewer 1 Report
Comments and Suggestions for Authors
Achounna et al. performed a pharmacological study to evaluate the effects compound EB1089 in combination of endocrine therapy or lapatinib in ER+ or ER- HER2+ breast cancer cell lines. There are some major concerns regarding to the experimental design and needs to be completed addressed.
1. The authors mainly claimed that EB1089 could sensitize the cells towards endocrine resistance or lapatinib to the two cell lines. However, they used single dose of drug treatment in all the experiments which could not sufficiently support their conclusionl The authros need to show the dose response curve shift of tamoxifen/fulvesntrant and lapatinib in the presence or absnece of combination and perform rigorous statistical test.
2. The study claims the differential effects on ER+ and ER- HER2+ breast cancer cell lines, but they only used one cell line from each group, which is not sufficient. They need to provide data from at least one more cell line from each group.
3. The western blot results always have uneven loading across the samples thus the change is not easy to be identified. The authors need to provide quantification of the protein and merge the data from at least three independent experiments plus statistic test to make the conclusions.
Comments on the Quality of English LanguageEnglish is fine. Minor edits is still needed.
Reviewer 2 Report
Comments and Suggestions for Authors
The manuscript titled "EB1089 increases the antiproliferative response of lapatinib in 2 combination with antiestrogens in HER2-positive breast cancer 3 cells" is a well-written experimental validation of the utilization of Seocalcitol drug in two cell lines modeling HER2+ breast cancer. The findings of this manuscript will have a translational impact if reproduced in animal models and in the drug-resistant derivatives of the cell lines used. But overall, the manuscript was a pleasure to read, and the data was well presented.
Reviewer 3 Report
Comments and Suggestions for Authors
In this manuscript, the authors showed the effect of EB1089 on BT474 and SKBP3 breast cancer cell lines. However, the effect of EB1089 was not consistent in these cell lines. The underlying mechanism explaining the difference is missing in the current submission, e,g why would EB1089 enhance ER expression in SKBP3 but not if BT474? Similarly, why do EB1089 affect pAKT and pSRC in BT474 but not in SKBP3? Much more work has to be done to uncover the mechanism. Validation in animal models is missing. This significantly compromises the quality of the manuscript.
1. How about the effect of EB1089 in the SKBP3 cell line in Figure 1?
2. The western blot result in figure 1 must be improved. The images in the BT474 panel had been cropped.
3. Apart from ERα, how about the effect of EB1089 on HER2 protein expression?
4. Regarding table 1, the response curves should be shown.
5. Also, the effect on cell cycle and apoptosis has remained unaddressed.
6. Based on the results from Figures 1, 4, 5, The effect of EB1089 would have a different effect on ERα, pAKT, pSRC in BT474 and SKBP3. It is difficult to conclude the effect of EB1089 in breast cancer, and it would be difficult to predict if a similar effect would be seen in other ER-negative/HER2-positive cell lines. The authors must prove that the effect can be recapitulated in a set of cell lines for a particular breast cancer subtype. In addition, the author did not determine if the HER2 signalling cascade would be affected.
7. Also, an in vivo model is absent, like a xenograft study. To confirm the therapeutic effect, an animal model is needed. However, unless the study focuses on explaining the molecular mechanism of EB1089 and its combined effect, this information seems missing in the current submission.
Round 2
Reviewer 1 Report
Comments and Suggestions for Authors
It appears 2/3 of my major concerns were not sufficiently addressed.
For question#1, the experiments asked was not for different doses of EB1089, but rather the dose respons for anti-estrogens and lapatinib. A dose response or IC50 shift is typically used to show the sensitivity change. Merely use single dose of anti-estrogens and lapatinib is not convincing as it could be at a plateua of the curve. So this experiment is still needed to consolidate the conclusion.
For question#2, usage of multiple cell lines in previous publication does not support the same conclusion for these cell lines in this study as each study tested a different questions. So the authors should either provide additional data from different cell lines or rephrase the text to limit the conclusion within these two cell lines rather than expanding them to entire ER+ and ER-HER2+ breast cancer.
Comments on the Quality of English LanguageEnglish quality is good.
Author Response
Reviewer 1
Comments and Suggestions for Authors
It appears 2/3 of my major concerns were not sufficiently addressed.
We extend our sincerest appreciation for your review of our manuscript and for
providing invaluable comments and suggestions to enhance its quality. We have
addressed the recommended revisions, highlighting all changes in yellow within the revised manuscript.
For question#1, the experiments asked was not for different doses of EB1089,
but rather the dose respons for anti-estrogens and lapatinib. A dose response
or IC50 shift is typically used to show the sensitivity change. Merely use single
dose of anti-estrogens and lapatinib is not convincing as it could be at a
plateua of the curve. So this experiment is still needed to consolidate the
conclusion.
In response to your valuable suggestion and our commitment to elevating the
manuscript's quality, we have requested a brief extension from IJMS to conduct the additional experiments as you recommended. Regrettably, our extension request
was declined as the proposed duration exceeded the standard revision time allotted by IJMS. Despite this setback, we genuinely appreciate your constructive comments, particularly emphasizing the significance of conducting dose-response curves for anti-estrogens and lapatinib to illustrate sensitivity changes. Recognizing the importance of this, our revised manuscript now incorporates the following paragraph: “Future experiments involving dose-response curves for tamoxifen, fulvestrant, and lapatinib should be performed to refine the experimental approach and explore a broader concentration range. This will provide a comprehensive understanding of sensitivity changes across various concentrations, ultimately contributing to reducing the concentration of antineoplastics”.
For question#2, usage of multiple cell lines in previous publication does not
support the same conclusion for these cell lines in this study as each study
tested a different questions. So the authors should either provide additional
data from different cell lines or rephrase the text to limit the conclusion within
these two cell lines rather than expanding them to entire ER+ and ER-HER2+
breast cancer.
We acknowledge the importance of clarity and specificity in our conclusions. In
response to this suggestion, we have revised the manuscript to limit the conclusion to the specific cell lines utilized in this study, namely BT-474 and SK-BR-3 HER2-positive breast cancer cells. We recognize the distinction between each study and concur that the conclusion should be tailored to the findings from these specific cell lines. This refinement will enhance the precision of our decisions and align more closely with the scope of this investigation.
In our revised manuscript, the conclusions have changed: “EB1089 enhances the
antiproliferative activity of endocrine treatment alone or in combination with lapatinib in BT-474 (ER-positive / HER2-positive) breast cancer cells. Furthermore, EB1089 restores antiestrogen responsiveness and increases the antineoplastic activity of the combined lapatinib treatment with antiestrogens in SKBR3 (ER-negative / HER2-positive) breast cancer cells. These effects are mediated through the modulation of ERα expression and the inhibition of Akt phosphorylation.

Reviewer 3 Report
Comments and Suggestions for Authors
The authors have addressed all the comments.
Author Response
Reviewer 3
Comments and Suggestion for Authors
The authors have addressed all the comments.
Dear Reviewer,
We appreciate your thougful evaluation or our manuscript. Your positive feedback is encouraging.

Round 3
Reviewer 1 Report
Comments and Suggestions for Authors
I think extend the revision period to add the dose response experimenst could be helpful to make these conclusion solid. Thus I would respecfully ask for the experiments to added in the next round of revision.
Comments on the Quality of English LanguageMinor editing of English language required.
Round 4
Reviewer 1 Report
Comments and Suggestions for Authors
The authors have addressed my remaining concern. I agree with the publication.